behaviour

body condition, behaviour, state-dependent decision-making, anti-predator behaviour, foraging, beaked whale

**Authors for correspondence:**
Eilidh Siegal
e-mail: eilidh.siegal@gmail.com
Patrick J. O. Miller
e-mail: pm29@st-andrews.ac.uk

# Beaked whales and state-dependent decision-making: how does body condition affect the trade-off between foraging and predator avoidance?

Eilidh Siegal, Sascha K. Hooker, Saana Isojunno and Patrick J. O. Miller

Sea Mammal Research Unit, Scottish Oceans Institute, University of St Andrews, St Andrews KY16 8LB, UK

ES, 0000-0002-7833-302X; SKH, 0000-0002-7518-3548; SI, 0000-0002-2212-2135

Body condition is central to how animals balance foraging with predator avoidance—a trade-off that fundamentally affects animal fitness. Animals in poor condition may accept greater predation risk to satisfy current foraging 'needs', while those in good condition may be more risk averse to protect future 'assets'. These state-dependent behavioural predictions can help interpret responses to human activities, but are little explored in marine animals. This study investigates the influence of body condition on how beaked whales trade-off foraging and predator avoidance. Body density (indicating lipid-energy stores) was estimated for 15 foraging northern bottlenose whales tagged near Jan Mayen, Norway. Composite indices of foraging (diving and echolocation clicks) and anti-predation (long ascents, non-foraging dives and silent periods reducing predator eavesdropping) were negatively related. Experimental sonar exposures led to decreased foraging and increased risk aversion, confirming a foraging/perceived safety trade-off. However, lower lipid stores were not related to a decrease in predator avoidance versus foraging, i.e. worse condition animals did not prioritize foraging. Individual differences (personalities) or reproductive context could offer alternative explanations for the observed state-behaviour relationships. This study provides evidence of foraging/predator-avoidance trade-offs in a marine top predator and demonstrates that animals in worse condition might not always take more risks.

## 1. Introduction

As behaviours that increase an animal's foraging efficiency often increase predation risk [1,2], animals must trade-off the risk of starvation against the risk of predation [3]. Starvation–predation trade-off theory predicts that the behavioural options of feeding animals lie on a continuum between energy maximization (at the complete expense of predator avoidance) and predation-risk minimization (at the complete expense of feeding) [3]. However, unless an animal is at imminent risk of starvation or predation, their optimal behaviour will likely fall somewhere in-between these two extremes, with animals balancing several aspects of their foraging behaviour (e.g. where, when, what and how to eat) with behaviours that avoid predators [3]. Body condition is expected to drive how animals balance these foraging versus predator-avoidance trade-offs: individuals in poor body condition (low energy reserves) are predicted to accept higher predation risk as they 'need' to increase foraging [4,5] and recover to better condition [6], while individuals in good condition (high energy reserves) are predicted to be more risk averse in order to protect their future assets—as in the 'asset-protection principle' [7,8].

Such needs- and assets-based explanations of how animals balance foraging behaviour against predation risk [5,7]—hence termed the needs/assets

hypothesis—are cornerstones of foraging theory (e.g. [8,9]) and can help understand the impacts of anthropogenic disturbance. If perceived by animals as analogous to predation risk, anthropogenic disturbance should increase avoidance behaviours at the cost of decreased foraging [10]. Body condition may alter this response with individuals in worse body condition (i.e. at greater starvation risk) less likely to adjust their behaviour following human impact [11], consistent with the needs/assets hypothesis [5,7].

Despite their importance for understanding foraging theory and how animals respond to human activities, condition-dependent behavioural predictions have been little explored in the marine environment [12–15]. Marine animals live within a dynamic fearscape, where they must balance heterogeneously distributed and often temporally fluctuating foraging benefits against variable predation risk [15,16]. From diel vertical migration of zooplankton (e.g. [17]) to refuge use by polychaete worms (e.g. [18]), marine organisms continually trade-off energy maximization for predation-risk minimization; yet, few studies have examined the extent to which these trade-offs are condition dependent. Empirical data so far have been consistent with the needs/assets hypothesis: starved intertidal gastropods accepted higher predation risk in order to increase foraging activity [14], green sea turtles (Chelonia mydas) in poor condition selected microhabitats with higher predation risk but more profitable foraging [13] and northern elephant seals (Mirounga angustirostris) rested more during the safety of night-time and at deeper depths as they gained lipid stores during feeding migrations [15].

Bio-logging technology provides a powerful tool for exploring condition-dependent risk-taking concepts in difficult-to-study marine animals [15]. As well as foraging [19] and anti-predator behaviours (e.g. [15,20]), these data-loggers can simultaneously provide information on the body condition of tagged mammals by using hydrodynamic models of buoyancy and drag forces on glide speeds to estimate body density (e.g. [21–24]). Low-body density indicates a high ratio of lipids to dense lean tissue and thus greater lipid-energy stores [21]. This method has been used to estimate the body density of northern bottlenose whales [23]—a member of the beaked whale (Ziphiidae) family. Although beaked whales can be difficult to study given their offshore habitats and routine dives to deeper than 800 m, they are a particular concern for anthropogenic disturbance, in part due to their sensitivity to naval sonar [25]. Sound- and movement-recording tags show that beaked whales perform foraging dives using echolocation clicks to search for, and buzzes to capture, deep-water squid and fish (e.g. [26]). These animal-borne data can also indicate their anti-predator behaviours: acoustic crypsis (silence) at shallow depths, synchronization of echolocation clicking within groups, bounce dives (a series of relatively shallow, silent dives in-between vocal foraging dives) and long-straight silent ascents from vocally active foraging dives (e.g. [20,27–30]); all of which may reduce the risk of visual and/or acoustic detection by shallower diving killer whales (Orcinus orca), their main predator [29].

Using such data-logging tags on northern bottlenose whales, we evaluated the trade-off between foraging and anti-predator behaviours and determined how this trade-off relates to body condition. We (i) developed composite indices of foraging and anti-predator indicators to investigate evidence of behavioural trade-offs between foraging and predator avoidance, (ii) used experimental exposures to human disturbance (sonar signals) to test whether the behavioural indicators were involved in trade-offs based on an increase in perceived risk and (iii) determined whether foraging versus predation-avoidance trade-offs were affected by body condition. Consistent with studies in marine animals [13–15], which support the needs/assets hypothesis [5,7], we a priori predicted that individuals in worse body condition should take more risks to satisfy their foraging needs, while those in better condition would take less risks in order to protect their future assets.

## 2. Material and methods

### (a) Data collection

Data were collected from sound- and movement-recording tags (DTags) deployed on 15 northern bottlenose whales north of Iceland, near the remote island of Jan Mayen, Norway in 2013–2016 (electronic supplementary material, S1). Animals were detected visually, or acoustically via a towed hydrophone array, and tagging was attempted opportunistically on adult-sized whales that approached the research vessel. Tags were deployed using a 5 m-long carbon-fibre hand-pole or a compressed air pneumatic launcher (Aerial Rocket Tag System, LK-ARTS) and attached via suction cups. Conductivity–temperature–depth (CTD) casts or temperature-only casts were made to estimate seawater density (used to fit the hydrodynamic models).

Tags sampled pressure, three-axis acceleration and three-axis magnetism at 50 Hz or 250 Hz. Sensors streams were decimated to 50 Hz and converted to whale-frame axes using established methods [31]. Audio was sampled at 192–240 kHz and acoustic recordings were audited manually by several experts. Aural cues and spectrogram visualization were used to identify regular clicking (long series, typically greater than 45 s, of regular clicks with interclick intervals typically between 0.2–0.4 s) and buzzes (fast click trains with interclick intervals approximately < 0.1 s). Although some inter-auditor variation in the identification of buzzes and regular clicking cannot be ruled out, all auditors followed a standardized protocol to minimize any inconsistencies.

Three of the 15 tagged whales were exposed to controlled sonar exposures [27,28]. These three individuals were included in all the data analyses. However, only baseline (pre-exposure) data were used to assess foraging versus predator-avoidance trade-offs and the relationship between these trade-offs and body condition. Post-sonar exposure data were used to test whether foraging/predator-avoidance trade-offs were based on an increase in perceived risk (i.e. sonar exposure).

### (b) Estimating body density

Body density ($\rho_{tissue}$) was estimated as in Miller et al. [23]. The method comprises (i) glide extraction and (ii) fitting measurements during gliding with a hydrodynamic performance model. Dorsal–ventral acceleration data were high-pass filtered (0.19–0.25 Hz) to reduce the gravitational components. Oscillation amplitudes below deployment-specific thresholds (0.1–0.5 m s$^{-2}$) were identified as glides. Depth-specific sea water density was estimated using the nearest-in-time CTD or temperature-only cast. During stable 5 s glide segments, measurements (acceleration, sea water density, speed and pitch angle) were fit to the hydrodynamic model to estimate body density using Bayesian methods [23]. Parameter-specific prior ranges and hierarchical model structures were set as in Miller et al. [23]. The Bayesian Markov chain Monte Carlo sampling method (three chains with 24 000 iterations, a 12 000-iteration burn-in, and a down-sampling factor of 36) was implemented using the coda [32] and R2jags R packages [33]. Trace histories and

Brooks–Gelman–Rubin diagnostic plots were used for model-fitting diagnostics and checking model convergence.

## (c) Behavioural indicators

Behavioural indicators were summarized for each baseline period for each tag deployment. The first 20 min of deployments were excluded to reduce the influence of any behavioural reaction to tagging. Individuals consistently returned to foraging—deep dives with buzzes—within 20 min after tagging.

### (i) Foraging indicators

Three behavioural indicators represented overall foraging effort: (i) overall buzz rate (buzz $h^{-1}$, a proxy for the rate of prey-capture attempts), (ii) the percentage of the deployment time spent producing regular clicks (representing search effort) and (iii) the percentage of time spent in foraging dives (representing foraging dive effort). Northern bottlenose whales typically exhibit either long and deep foraging dives or short and shallow non-foraging dives [27,34]. Dive types were thus identified via $K$-means clustering based on dive duration, maximum depth, descent and ascent rates [27,34], for dives deeper than 10 m. The silhouette coefficient [35] indicated that three dive types (short-shallow, mid-depth and long-deep) provided optimal categorization. Mid-depth and long-deep dives containing regular clicks were considered foraging dives. Consistent with foraging in deep-diving odontocetes (e.g. [36]), these dives had long bottom-phase durations, and contained buzzes and buzz-associated sudden movements.

### (ii) Anti-predator indicators

Five anti-predator indicators were estimated based on the anti-predator behaviours suggested for beaked whales (table 1). Using data from the ascents of foraging dives, we calculated: (i) mean ascent-pitch-shallowness (the inverse [90° - pitch] of the mean pitch during ascents from foraging dives), (ii) the mean depth at which silent ascents started (cessation-of-clicking depth) and (iii) the ascent-straightness index estimated as $1 - (STL - DMG)/STL$, where $STL$ = stretched-out track length of the ascent and $DMG$ = the distance made good (the distance actually covered in the ascent), where $0$ = extreme tortuosity and $1$ = completely straight ascent [39] (table 1). Ascents were considered the time from when regular clicking ceased (i.e. the time at which animals were last broadcasting their location to a potential predator) until the time the animal reached the surface [30]. (iv) We calculated the percentage of time near the surface (less than 20 m depth) spent not vocalizing, termed 'surface silence'. (v) We determined the proportion of non-foraging time (between foraging dives) spent in silent (bounce) dives deeper than 70 m (table 1).

### (iii) Composite indices of foraging and predator avoidance

To estimate an overall level of foraging and anti-predator behaviour for each individual, the foraging and anti-predator indicators were incorporated into two composite indices. Composite indices combine several potentially correlated metrics into a single index [40]. Based on sociality indices by Sapolsky et al. [41] and Silk et al. [40], the composite foraging index ($CI_F$) and composite anti-predator index ($CI_{AP}$) were calculated for each deployment as

$$CI = \frac{\sum_{i=1}^{n}(x_i/m_i)}{n},$$

where $n$ is the number of behavioural indicators, $x_i$ is the deployment-specific value for each indicator and $m_i$ is the median value of the indicator across all deployments. This index quantifies the

deviation of an individual from the population average across all metrics combined [40,41].

## (d) Statistical analyses

### (i) Foraging versus predator-avoidance trade-off

To examine evidence for a trade-off between foraging and anti-predator behaviour, $CI_{AP}$ was modelled as a function of $CI_F$ using a linear model with a Gaussian error distribution and identity link function. Model assumptions were checked using residual plots (electronic supplementary material, S2), and the presence of influential observations was assessed with Cook's distance. We predicted $CI_{AP}$ to decrease as $CI_F$ increased, following a foraging versus predator-avoidance trade-off [3]. Spearman's rank correlation coefficient was estimated between each foraging and anti-predator indicator. Mean values are reported alongside the standard error of the mean.

### (ii) Foraging and anti-predator behaviour under increased perceived risk

Sonar signals are likely perceived as a threat by northern bottlenose whales [27,28]. Changes in foraging and anti-predator behaviours before and after controlled sonar exposures ($n = 3$) were therefore compared to test whether these behavioural indicators were altered by increased perceived risk. We predicted that anti-predator indicators should increase after sonar exposure, while foraging indicators should decrease. Post-exposure periods were calculated over the time from the end of the sonar exposure to the end of the deployment.

The time regular clicking ceased was a key component of the predator-avoidance behaviours and was used to define the start of the ascent phase (table 1). However, following sonar exposures, there was no (or only limited) regular clicking. Consequently, for comparing post-exposure periods to pre-exposure baseline data, three anti-predator indicators (the ascent start depth, ascent-pitch-shallowness and the ascent-straightness index) were re-calculated for all mid-depth and long-deep dives, with the start of the ascent phase defined as the last time an animal's pitch was less than 0° [22]. The composite anti-predator index estimated using this pitch definition was termed $CI_{AP\_pitch}$.

### (iii) Influence of body condition on foraging versus anti-predator behaviour

To examine the relationship of the behavioural indices to body condition, the foraging–predation risk trade-off was presented as a ratio, $CI_{AP}/CI_F$ (e.g. [42]), which was modelled as a function of body density. As predicted by condition-dependent risk-taking theory, individuals with higher body density (worse condition) were expected to have a lower ratio of predator avoidance to foraging. General linear models (with a Gaussian error distribution and identity link function, electronic supplementary material, S2) with different covariates were compared using Akaike information criterion with a correction for small sample sizes (AICc).

Group size, deployment duration and dominant stroke frequency were included as covariates. Group size at the time of tagging was included as social context can affect anti-predator behaviours [3]. Deployment duration was included as deployment durations ranged widely (1.2–12.9 h) and despite exclusion of the first 20 min, shorter duration deployments might contain relatively more avoidance behaviour induced by the tagging process [43]. As animal size can potentially affect body density [23], dominant stroke frequency, which is proportional to animal size [44], was included as an interaction term with $\rho_{tissue}$. Spearman's rank correlation coefficient between

**Table 1.** Behavioural indicators used to define anti-predator behaviour in beaked whales. These behaviours are predicted to reduce the risk of being visually and/or acoustically detected by shallow-diving killer whales.

| behaviour | description and justification | behavioural indicator | example species |
|---|---|---|---|
| long shallow-pitched ascents | prolonged ascents from foraging dives with low pitch angle and vertical speed [29,30]. Ascending at a low pitch increases the time and horizontal displacement from the cessation-of-clicking depth to re-surfacing; thus, increasing uncertainty in the prey's position for a stalking predator and decreasing predation risk [29] | ascent-pitch-shallowness (the inverse [90° - mean pitch] of the mean pitch during ascents from foraging dives) | *Hyperoodon ampullatus* [27,34]<br>*Mesoplodon densirostris* (e.g. [29,30])<br>*Ziphius cavirostris* (e.g. [29,30]) |
| acoustic crypsis—during ascents | cessation of clicking during ascents from foraging dives [29,30]. Ascending silently from foraging dives reduces the risk of being acoustically detected by near-surface predators. Individuals that cease clicking at deeper depths effectively broadcast their location further from potential predators [29], decreasing predation risk | mean depth that silent ascent commences from foraging dives (cessation-of-clicking depth) | *Hyperoodon ampullatus* [27]<br>*Mesoplodon densirostris* (e.g. [29,30,37])<br>*Ziphius cavirostris* (e.g. [29,30]) |
| horizontal displacement | consistent straight heading during ascents from foraging dives [29,30]. Ascending from foraging dives with high horizontal displacement (high straightness index) maximizes the horizontal separation from the cessation-of-clicking depth to the surfacing location, decreasing predation risk [29] | ascent-straightness index. Ranges from 0 (highly tortuous movement) to 1 (straight-line movement) | *Mesoplodon densirostris* (e.g. [29,30])<br>*Ziphius cavirostris* (e.g. [29,30]) |
| acoustic crypsis—at the surface | acoustic silence at shallow depths. Spending less time vocalizing near the surface decreases the risk of acoustic detection by near-surface predators [29] | % surface silence (percentage of time near the surface, <20 m, spent not vocalizing) | *Mesoplodon densirostris* (e.g. [29,37])<br>*Ziphius cavirostris* (e.g. [20,29]) |
| bounce dives | silent dives occurring in-between foraging dives. Silent dives to depths deeper than typical inter-ventilation depths reduce the risk of visual and acoustic detection by near-surface predators [15,30,38] | % silent dives (percentage of time between foraging dives spent in silent dives >70 m depth) | *Mesoplodon densirostris* (e.g. [29,30,38])<br>*Ziphius cavirostris* (e.g. [29,30]) |

body density and each behavioural indicator was calculated to further explore behaviour–body condition relationships.

## 3. Results

### (a) Behavioural indicators and the foraging versus predator-avoidance trade-off

Foraging indicators were generally positively correlated with each other and negatively correlated with anti-predator behaviour (table 2). As the three foraging indicators increased, the indicator of silent dives greater than 70 m decreased (table 2). An increase in foraging-dive effort correlated with a decrease in mean ascent-pitch-shallowness and the ascent-straightness index (table 2). An increase in $CI_F$ predicted a decrease in $CI_{AP}$ (slope = −0.48, $t_{13}$ = −3.99, $p$ = 0.002;

figure 1*a*). Approximately 55% of the variance in $CI_{AP}$ was explained by $CI_F$ ($R^2$ = 0.55). Cook's distance estimates (0.0– 0.2) indicated that no single datapoint overly influenced the model of $CI_{AP}$ versus $CI_F$ [45].

### (b) Foraging and anti-predator behaviour under increased perceived risk

Two of the three deployments exposed to experimental sonar had values of zero for all foraging indicators after exposures, while for the third deployment, foraging indicators were less than a third of the values before the exposures (figure 2).

After sonar exposures, anti-predator indicators predominantly increased (i.e. changed in the direction predicted to reduce predation risk; figure 2). The percentage of time spent being non-vocal near the surface (surface silence) increased

**Table 2.** Behavioural indicators of foraging and anti-predator behaviour, and body condition for 15 northern bottlenose whales tagged with sound- and movement-recording devices around Jan Mayen, Norway. Spearman's rank correlations between indicators, and summary values (mean ± standard error of the mean, s.e.m. and range) are shown for each indicator.

| | foraging indicators | | | anti-predator indicators | | | | | body condition |
|---|---|---|---|---|---|---|---|---|---|
| | buzz rate | % regular clicking | % foraging dive | ascent depth | ascent-pitch-shallowness (90° - ascent-pitch) | ascent-straightness index | % surface silence | % silent dives | $\rho_{tissue}$ |
| **foraging indicators** | | | | | | | | | |
| buzz rate | — | | | | | | | | |
| % regular clicking | 0.61* | — | | | | | | | |
| % foraging dive | 0.81*** | 0.71** | — | | | | | | |
| **anti-predator indicators** | | | | | | | | | |
| ascent depth | −0.06 | −0.36 | 0.00 | — | | | | | |
| ascent-pitch-shallowness | −0.54* | −0.34 | −0.63* | −0.35 | — | | | | |
| ascent-straightness index | −0.41 | −0.27 | −0.53* | 0.13 | 0.36 | — | | | |
| % surface silence | −0.24 | −0.18 | −0.11 | 0.01 | 0.42 | −0.08 | — | | |
| % silent dives | −0.57* | −0.90*** | −0.64* | 0.59* | 0.35 | 0.33 | 0.27 | — | |
| **body condition** | | | | | | | | | |
| $\rho_{tissue}$ | −0.48 | −0.44 | −0.16 | 0.33 | 0.06 | 0.24 | 0.14 | 0.48 | |
| **summary values** | | | | | | | | | |
| mean (± s.e.m.) | 17.2 buzz h⁻¹ (± 1.5) | 34.7% (± 2.9) | 46.2% (± 3.7) | 368.6 m (± 29.7) | 62.1° (± 2.2) | 0.6 (± 0.0) | 98.7% (± 0.4) | 41.7% (± 7.6) | 1030.8 kg m⁻³ (± 0.4) |
| range | 7.9–26.8 buzz h⁻¹ | 18.0–57.5% | 27.7–72.3% | 106.1–520.1 m | 40.6–76.5° | 0.4–0.9 | 94.1–100.0% | 0.0–86.3% | 1028.4–1033.9 kg m⁻³ |

$*p < 0.05$; $**p < 0.01$; $***p < 0.001$.

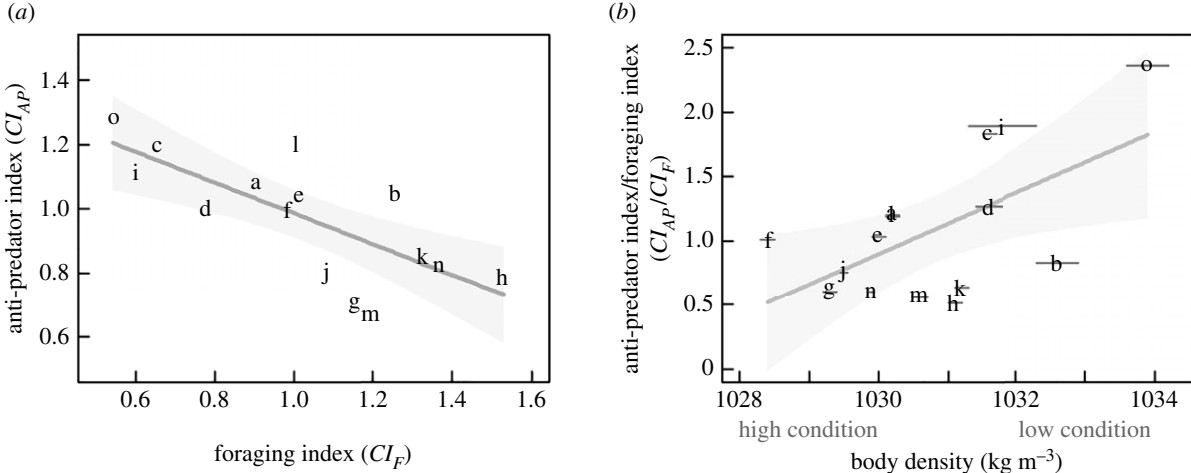

**Figure 1.** (a) The composite anti-predator index ($CI_{AP}$) as a function of the composite foraging index ($CI_F$), and (b) the ratio of the composite indices as a function of body density (± 95% posterior credible interval shown as horizontal lines), for 15 northern bottlenose whales (whale codes in electronic supplementary material, table S1) tagged with sound- and movement-recording devices around Jan Mayen, Norway. For the three individuals that were experimentally exposed to sonar (codes a, l and n), only baseline data were analysed. Predicted linear models (Gaussian error distribution and identity link function) are shown with the 95% confidence intervals (shaded). One deployment (code 'o') had a high influence (Cook's distance = 0.8) on the $CI_{AP}/CI_F$ versus body density model.

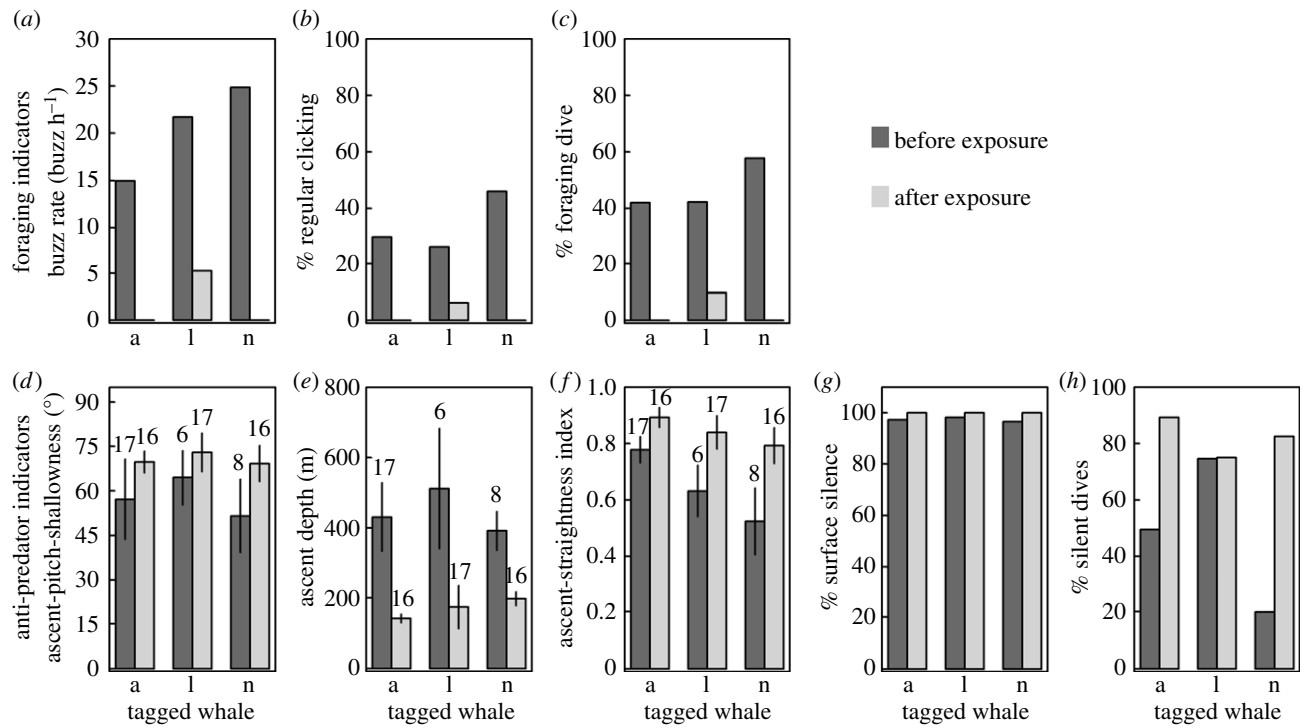

**Figure 2.** Behavioural indicators of foraging (top row) and predator avoidance (bottom row) before sonar exposures (5.0–10.6 h of baseline data) and after sonar exposures (7.5–9.5 h of post-exposure data) for three northern bottlenose whales (codes a, l and n) tagged with sound- and movement-recording devices around Jan Mayen, Norway. For ascent-pitch-shallowness, ascent depth and the ascent-straightness index, the error bars represent the standard error of the mean and the values above the bars represent the number of dives from which the mean values were obtained.

from 97.3% (± 0.5, n = 3, range = 96.5–98.1%) to approximately 100% (range = 99.9995–100%) and the indicator of silent dives (the percentage of time between foraging dives spent in silent dives greater than 70 m) increased by 0.4–62.6% (figure 2). Although the ascent phase started at a shallower average depth after exposures, animals generally ascended from mid-depth and long-deep dives with a shallower pitch and a higher straightness index (figure 2).

As a result of these changes in the behavioural indicators, $CI_F$ on average decreased (by 1.0 ± 0.2, n = 3) and $CI_{AP\_pitch}$ increased (by 0.1 ± 0.1, n = 3) following sonar exposures (electronic supplementary material, S1).

### (c) Influence of body condition on foraging versus anti-predator behaviour

Individual body density estimates obtained from the lowest DIC model (electronic supplementary material, S3) ranged from 1028.4 to 1033.9 kg m$^{-3}$ (mean = 1030.8 kg m$^{-3}$ ± 0.4, n = 15). As in Miller *et al*. [23], the model with the lowest deviance information criterion allowed for individual-average (global) plus inter-individual variation in body density and the combined drag term, as well as dive-by-dive variability in diving gas volume (electronic supplementary material, S3). Average deployment duration was 8.1 h (± 0.9, n = 15,

range = 1.2–12.9 h), mean dominant stroke frequency (a proxy for body size) was 0.3 Hz (± 0.0, $n = 15$, range = 0.1–0.5 Hz) and mean group size at the time of tagging was 4.1 (± 0.5, $n = 15$, range = 2–8).

Of the 12 general linear models tested, there was substantial support for $\rho_{tissue}$ as the sole predictor of $CI_{AP}/CI_F$ (electronic supplementary material, S4). With a Cook's distance of 0.8 [45], one individual likely had a large influence on the model of $CI_{AP}/CI_F$ as a function of body density (marked as 'o' in figure 1). This animal was an extreme case as it had the highest body density (1033.9 kg m$^{-3}$), low foraging indicators (buzz rate = 7.9 buzzes h$^{-1}$, search effort = 18.0%, foraging-dive effort = 27.7%) and strong anti-predator indicators (mean ascent depth = 380.0 m ± 43.2, $n = 5$; mean ascent-pitch-shallowness = 69.9° ± 3.2, $n = 5$; mean ascent-straightness index = 0.9 ± 0.0, $n = 5$; surface silence = 98.6%; indicator of silent dives greater than 70 m = 86.3%). When this animal was excluded, $CI_{AP}/CI_F$ did not vary with body density (slope = 0.13, $t_{12} = 1.23$, $p = 0.24$, $R^2 = 0.11$), indicating that there was no relationship between $CI_{AP}/CI_F$ and body density; consequently, animals in worse condition did not have higher $CI_F$ and lower $CI_{AP}$ (as expected by the needs/assets hypothesis). When this animal was included, the relationship between $CI_{AP}/CI_F$ and body density was positive (slope = 0.24, $t_{13} = 2.67$, $p = 0.02$, $R^2 = 0.36$), contrasting the negative relationship predicted *a priori*. Other covariates had little AIC support (electronic supplementary material, S4). The ratio ($CI_{AP}/CI_F$) did not vary with deployment duration ($t_{13} = 2.13$, $p = 0.05$), group size ($t_{12} = 0.53$, $p = 0.61$) or dominant stroke frequency ($t_{12} = 0.54$, $p = 0.60$). Body density did not correlate with any behavioural indicator (table 2). Neither $CI_F$ ($t_{13} = -1.42$, $p = 0.18$, $R^2 = 0.14$) nor $CI_{AP}$ ($t_{13} = 2.07$, $p = 0.06$, $R^2 = 0.25$) varied significantly with body density.

## 4. Discussion

Although a near universal challenge for animals [1], the foraging versus predator-avoidance trade-off and the influence of body condition on this trade-off have been little explored in marine animals in the upper trophic levels [15,46]. This study provides evidence of (i) a marine top predator trading off foraging for anti-predator behaviour (during apparently undisturbed periods), (ii) disturbance from sonar exposures corroborating this trade-off, decreasing foraging and increasing anti-predator behaviours and (iii) body condition not having the expected relationship with foraging versus anti-predation that had been *a priori* predicted from the needs/assets hypothesis of behaviour.

### (a) Evidence for the foraging versus predator-avoidance trade-off

Northern bottlenose whales appear to trade-off foraging effort for behaviours predicted to reduce predation risk (table 1), as proposed by starvation–predation trade-off theory [1,3]. An increase in the overall level of foraging ($CI_F$) was related to a decrease in the anti-predator index ($CI_{AP}$; figure 1). Correlations between individual behavioural indicators were also consistent with the expected foraging–predation-risk trade-off (table 2). For example, a decrease in the indicator of silent dives greater than 70 m (predicted to increase predation risk) [30] correlated with increased echolocation search effort, foraging-dive effort and the overall rate of prey-capture attempts (table 2).

The diving behaviour of marine mammals is driven by physiological constraints, as well as ecological factors, such as predation risk [47]. Balancing resource exploitation at depth with oxygen intake at the surface is a clear physiological constraint on their behaviour [47]. Animals that spent more time in foraging dives ascended from these dives at a steeper pitch (table 2), potentially to replenish oxygen stores sooner. However, given that beaked whales routinely ascend at shallow angles (table 2; e.g. [30]), this behaviour likely also reflects ecological needs, such as reducing predation risk [29,30]. Animals that spent more time in foraging dives had a lower indicator of silent dives (table 2), suggesting that these silent dives may have an anti-predator function rather than (or in addition to) previously proposed nitrogen-management functions [38]. Therefore, although physiological trade-offs are clear constraints on marine mammal behaviour, balancing foraging gains with predation risk appears to explain the behavioural patterns found in this study (figure 1 and table 2).

Although our behavioural measures were designed to limit inherent time trade-offs, some were inevitable given that foraging indicators involved producing echolocation sounds and anti-predator indicators involved being silent (table 1). One of the clearest examples of a time-budget trade-off in the context of the foraging versus predator-avoidance trade-off is the use of refuges from predators: the more time spent in refuges, the less time is available for foraging (e.g. [18]). For beaked whales, silence is an 'acoustic refuge' that reduces predation risk at the cost of echolocation-based foraging [29]. Controlling the time allocated to acoustic crypsis versus vocal foraging is therefore likely fundamental to how beaked whales manage the challenge of balancing foraging with predator avoidance [29].

### (b) Experimental sonar playback validates the foraging–predation-risk trade-off

Underlying the observed foraging versus predation-risk trade-off is the fundamental assumption that the selected anti-predator behaviours evolved to reduce predation risk; yet, this study did not quantify actual predation risk. Exposure to sonar sounds provided an opportunity to evaluate whether the anti-predator indicators played their expected role in mitigating perceived risk (table 1). Following sonar exposures, animals effectively ceased near-surface vocalizations, the indicator of silent dives greater than 70 m increased and whales ascended from dives with straighter shallower pitched ascents (figure 2), all suggesting increased risk aversion [29]. Although ascents began at shallower depths following exposures (figure 2), most post-exposure dives were silent. This shallower ascent depth therefore likely indicates that whales prioritized horizontal-avoidance behaviour [28]. Anti-predator indicators generally moved in the direction hypothesized to reduce predation risk after sonar exposures (figure 2), suggesting that these indicators (and the behaviours they represent, table 1) are involved in how northern bottlenose whales behaviourally control risk, although the fitness consequences of these anti-predator behaviours remain unclear.

## (c) Unexpected relationship between body condition and the foraging/anti-predator trade-off

According to the needs/assets hypothesis, animals in poorer condition should prioritize foraging at the expense of accepting increased predation risk [4], while those in better condition are expected to prioritize safety over energy maximization [15]. While body density was supported as a predictor of the anti-predator to foraging index ratio (electronic supplementary material, S4), one deployment (labelled 'o' in figure 1) strongly influenced the model fit, so that when this one animal was excluded, there was no statistical relationship between $CI_{AP}/CI_F$ and body density. When this animal was included, higher body density (worse body condition) correlated with higher $CI_{AP}/CI_F$ (higher levels of predator-avoidance behaviour relative to lower levels of forging). However, given the strong dependence of this relationship on the one animal (though we have no reason to doubt the validity of the datapoint), and that this study is cross-sectional rather than longitudinal (i.e. the data represent a snapshot of behaviour at one point in time), we do not conclude that body density is positively related to $CI_{AP}/CI_F$. Ultimately, however, whether this data point was included or not, the results did not support a *negative* relationship between body density and $CI_{AP}/CI_F$ as was expected by the needs/assets hypothesis: i.e. animals in worse body condition did not have lower levels of anti-predator behaviour and higher levels of foraging.

One explanation for these results could be that the range of observed body conditions might not have been great enough for body condition to drive increased risk-taking behaviour in poorer condition animals [7]. Across the 15 individuals, body density varied by 5.5 kg m$^{-3}$ (figure 1)—a small range compared to humpback whales (*Megaptera novaeangliae*), which, with seasonal migrations, experience vast differences in their lipid stores across seasons [24]. The migratory patterns of bottlenose whales are unclear, but there is little evidence of fasting–feasting cycles [48]. Lower body density (fatter) bottlenose whales were found around Jan Mayen compared to those in the northwest Atlantic [23]. If Jan Mayen individuals were above their starvation risk threshold (i.e. they had fasting endurance), and close to their reproductive threshold (i.e. they had sufficient resources for successful reproduction), they may be more cautious, prioritizing survival from predation [49]. Additionally, beaked whales may generally be constrained to maintain narrow body-fat limits, as even small changes to their volumes of wax-ester blubber could impact their ability to dive efficiently to deep depths [48,50]; thus, differences in body density across individuals may not be enough to drive risk-taking behaviour [5].

An alternative to the needs/assets hypothesis, the state-dependent safety hypothesis (an 'ability-based' explanation; [5]) could also influence state-behaviour trade-offs. This hypothesis predicts that individuals in better condition may take greater predation risks due to having abilities or traits that increase the probability of a successful outcome [5,8,51]. For instance, if animals in better condition can flee faster or defend themselves better, they may take more risks [5], such as being more conspicuous to predators [51]. Beaked whales show deep dives and extreme horizontal movement responses following exposure to killer whale and mid-frequency sonar sounds (e.g. [27,52]). In this study, animals in better condition (lower body density) were likely to be closer to neutral buoyancy [23], and since energetic cost of movement (cost-of-transport) is reduced at neutral buoyancy [50], these higher condition individuals may have greater ability to flee, allowing greater risk-taking. However, the results did not provide strong evidence that higher condition animals could exhibit more risk-prone behaviour (figure 1b). Consequently, similarly to the needs/assets hypothesis, inter-individual differences in body condition might not have been extreme enough to impact an individual's ability to avoid predation and thus drive the state-dependent safety hypothesis.

Individual differences in behaviour that persist across contexts and time—termed 'personalities'—could also shape the foraging versus predator-avoidance trade-off [53]. Increased risk-taking through one anti-predator behaviour (cessation-of-clicking depth) equated to increased risk-taking in another (the indicator of silent dives greater than 70 m), suggesting that there was consistency in risk-taking behaviour for particular animals (tables 1 and 2). The backbone of animal personalities or behavioural syndromes is that individuals routinely exhibit above or below average differences in behaviour across different contexts and over time [53]. Bolder or proactive risk-taking individuals may forage more efficiently and consequently have better body condition [51]. From such feedback loops wherein personality has fitness consequences, a positive relationship between body density and $CI_{AP}/CI_F$ would be expected as individuals with risk-averse foraging strategies (higher $CI_{AP}/CI_F$) might prioritize increased survival over foraging, and thus obtain less energy and have worse body condition (higher body density). This could be the case for the one extreme deployment (labelled 'o' in figure 1), which had the poorest body condition, high levels of anti-predator behaviour and low levels of foraging. However, when excluding this individual datapoint, the results did not provide clear evidence that risk-averse animals had lower body condition (figure 1b). To determine the presence of personalities, repeated measures of individual behaviour over time (a longitudinal study) are required. Consistency observed for the 1 to 13 h time period of this study (electronic supplementary material, S1) is suggestive, but could simply reflect the environmental factors or context experienced by the individual over the course of the tag deployment. It is possible that close approaches required for tag deployment may unintentionally select animals with bolder and/or more risk-prone personalities [54], resulting in less plasticity in foraging and risk-taking behaviours [55]. The lack of relationship between body density and $CI_{AP}/CI_F$ could therefore be due to this study being biased towards bolder individuals that did not demonstrate behavioural plasticity in response to body condition as had been predicted *a priori* (figure 1b); however, a longitudinal study would be required to confirm this.

Life-history context, social context, temporal scale and prior history may also be relevant. The asset-protection principle predicts that individuals with larger current reproductive assets will be more risk averse; thus, pregnant females and/or females with offspring may be less prone to take risks [7]. In addition to sex and age (e.g. [46]), group composition may also affect behavioural trade-offs, particularly as beaked whales exhibit highly synchronized anti-predator and foraging behaviours [29]. Furthermore, while body density may be the outcome of several days–months of energy intake and consumption (e.g. [21]), deployments in this study averaged 8 h in duration. Short-term contextual factors (e.g. behavioural state, prey availability and risk perception) are therefore likely more relevant at this scale. We also do not know the history

of the tagged animals. It is possible that the individual with the extreme characteristics (high body density and high anti-predator behaviours, labelled 'o' in figure 1) might have been in the area and exposed to sonar 3 days previously, potentially causing sensitization in its high anti-predator behaviour (electronic supplementary material, S1).

## 5. Conclusion

Human disturbance, such as sonar exposure (figure 2), increases safety-seeking behaviour at the expense of other fitness-enhancing activities [10]. Northern bottlenose whales appear to exchange foraging for behaviours involved in managing predation risk, as predicted by starvation-predation trade-off theory [3]. Although body condition was expected to drive these foraging/predator-avoidance trade-offs [4] through the needs/assets hypothesis (animals in poor condition increase risk-taking so as to increase their condition, while better-condition animals are more risk-averse to protect their future assets) [5,7], the empirical results of our study did not support this. Worse body condition was not associated with increased risk-taking and higher levels of foraging (figure 1*b*) as expected by the needs/assets hypothesis. However, support for a positive relationship between risk-taking behaviour and body condition was limited due to the influential role of one datapoint. With slow life-history strategies [56], even when in poor condition, beaked whales may be unwilling to increase predation risk, choosing instead to preserve their ability to reproduce in the future—in essence choosing 'life' over 'dinner' [57]. It is possible that these long-lived animals will only take more risks when close to starvation or will decrease foraging only when fat stores become excessive and reduce manoeuvrability [12]. This study provides a basis

for simultaneously assessing different behaviours (foraging and anti-predation) and their relationship with body condition via non-invasive methods in a difficult-to-study marine predator. We demonstrate that state-dependent behavioural predictions might not always hold true for animals whose body condition is perhaps not extreme enough to drive relationships between body condition and behavioural trade-offs.

Ethics. The research was approved by the University of St Andrews Animal Welfare and Ethics Committee. Tagging and sonar exposure experiments were conducted under permits from the Icelandic Ministry of Fisheries and the Norwegian Animal Research Authority (permit numbers 2011/38782 and 2015/23222).

Data accessibility. The dataset supporting this article are available from Dryad Digital Repository: https://doi.org/10.5061/dryad.qrfj6q5hj [58].
The data are provided in the electronic supplementary material [59].

Authors' contributions. E.S.: conceptualization, data curation, formal analysis, investigation, methodology, visualization, writing—original draft and writing—review and editing; S.K.H.: conceptualization, methodology, supervision, writing—review and editing; S.I.: data curation, methodology, writing—review and editing; P.J.O.M.: conceptualization, data curation, funding acquisition, methodology, project administration, resources, supervision and writing—review and editing.
All authors gave final approval for publication and agreed to be held accountable for the work performed therein.

Competing interests. The authors declare that they have no competing interests.

Funding. This work was supported by the US Strategic Environmental Research and Development Program (award RC-2337), the US Office of Naval Research and the French DGA.

Acknowledgements. Thanks to all field teams involved in data collection around Jan Mayen, to C. Curé for providing playback equipment, and to L. Martín López and N. Biassoni for scoring acoustic recordings. Thank you to L. Rendell, V. Deecke, the University of St Andrews Behaviour Discussion Group, and two anonymous reviewers for valuable advice.

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
