## [Peer Review File · Proceedings of the Royal Society B: Biological Sciences]

Review History

RSPB-2021-1160.R0 (Original submission)

Review form: Reviewer 1

Recommendation

Reject – article is scientifically unsound

Scientific importance: Is the manuscript an original and important contribution to its field?

Good

General interest: Is the paper of sufficient general interest?

Good

Quality of the paper: Is the overall quality of the paper suitable?

Good

Is the length of the paper justified?

Yes

Should the paper be seen by a specialist statistical reviewer?

No

Do you have any concerns about statistical analyses in this paper? If so, please specify them explicitly in your report.

No

It is a condition of publication that authors make their supporting data, code and materials available - either as supplementary material or hosted in an external repository. Please rate, if applicable, the supporting data on the following criteria.

Is it accessible?

No

Is it clear?

No

Is it adequate?

No

Do you have any ethical concerns with this paper?

No

Comments to the Author

I found this to be an extremely well written paper on an interesting topic. The introduction was strongly rooted in theory, and beaked whales provide an interesting case study for answering questions about state-dependence given the relative ease of calculating foraging and predator-avoidance behaviors as well as body condition. The statistical analysis was generally strong, and the authors were creative in their use of anti-predator and foraging metrics. However, I struggled with the interpretation of the main data figure (specifically, the leverage that one datapoint has on the model fit and thus interpretation and implications of the paper). I am concerned that given the cross-sectional rather than longitudinal approach, and the strong impact of a single datapoint that could easily have been affected by personality or habituation, question the strong statements underlying the pitch of the paper. As a result, I feel that the language needs to be softened throughout the paper, and that the manuscript may be better suited to a slightly lower tiered journal.

Major comments

- L26-28 suggests that skinnier animals minimized predation risk... But this is a chicken-or-egg question (what came first?). What if body condition was low because animals used more anti-predator behaviors at the expense of foraging? I don't think this necessarily suggests that the needs/assets hypothesis is not true.
- When the one animal was excluded, CLAP/CLF did not vary with body density - so statements in the abstract should be softened. Figure 1b should not have a line because it is not significant. L25-28 is not supported by the data. Without this result, the novelty of the manuscript is somewhat reduced (because, for example Figure 1A - the negative relationship between foraging and anti-predator behaviors - are an emergent property, as discussed in L310). Also, this should not be interpreted as strong negative support for the needs/assets hypothesis, as is written into several sections of the paper (e.g., L284-286).
- Much of this work, especially the discussion, is pitched as the starvation-predation trade-off. While I appreciate this effort, I think the tradeoff being measured here is foraging versus predator avoidance. The authors could add a few sentences to discuss this difference. For example, what are the body density values in this species that might lead to starvation?

Minor comments

Abstract: In abstract, response to sonar exposure is termed predator avoidance... should this be considered risk aversion? Threat avoidance? I agree it is analogous, but I do not think that behavioral changes from sonar exposure should be called predator avoidance. I appreciate the nuanced discussion section (e.g., L330-334) and would like to see more careful wording in the

abstract.

L13: Start to abstract is dry

L18: grammar

L20: Where foraged, not tagged

L21: Is acoustic crypsis an anti-prey behavior as well?

L25: Simplify to lower lipid stores (remove higher body density)

L30: Abstract concluding sentence should broaden out results / discuss implications rather than justify why a certain result was not found

L65: The safety component of this should be underscored. Suggest changing to “rested more during the safety of nighttime and at deeper depths...”

L75: Suggest giving an example of anthropogenic disturbances that are concerning. Sonar?

L86: Change “this study” to “we”

L88: Suggest removing “of this marine top predator”

L90: It was unclear to me in the abstract whether sonar was experimentally added or whether it was naturally encountered. Here it is clear that experimental exposures were done – please add this to the abstract.

L124: Change grammar to “During stable five-second glide segments, measurements (acceleration...)”

L139: What is the unit for buzz rate?

L148: What constitutes “regular clicks”? Is this objective or subjective?

L149: Change to “long bottom phase durations”

L151: I found this paragraph relatively hard to follow. Perhaps a figure would help (i.e., convert the table to a figure with more helpful information).

L234: I would like to see statistical analysis in this section. Are these comparisons biologically or statistically relevant? For example, the percentage of time being non-vocal near the surface increased from 97.3 to 100%, which is a very minor change.

L213: Is there precedent for using dominant stroke frequency as an index of body size? I see in L218-219 that this is repeated, with citations, and now understand that L213 is meant to be an overview of the paragraph. I suggest removing “(an index of body size)” in L213 to resolve this confusion.

L258: I don’t believe it is necessary to specify that delta AICc is 0 here – this is clearly the best fitting model.

Fig 2: This is somewhat unclear to me – are 2013, 2015, and 2016 the three individual whales, or years, or both? Also, it would be helpful if the foraging metrics (presumably A,B,C) are somehow differentiated from the other metrics. For this figure and the others, it would be helpful to have more information about how many dives are represented.

L290: Replace “predicted” with “was related to”

L352: Replace “which annually cycle their lipid stores” with something like “which experience vast differences in lipid stores across seasons”

L377: This paragraph epitomizes my struggle with the inclusion of individual O in the regression. Because this study is cross-sectional rather than longitudinal, it seems that one whale with high (low) body condition could have simply been less of a risk taker, and has strong leverage on the linear model fit.

Review form: Reviewer 2

Recommendation

Accept with minor revision (please list in comments)

Scientific importance: Is the manuscript an original and important contribution to its field?

Excellent

General interest: Is the paper of sufficient general interest?

Excellent

Quality of the paper: Is the overall quality of the paper suitable?

Excellent

Is the length of the paper justified?

Yes

Should the paper be seen by a specialist statistical reviewer?

No

Do you have any concerns about statistical analyses in this paper? If so, please specify them explicitly in your report.

No

It is a condition of publication that authors make their supporting data, code and materials available - either as supplementary material or hosted in an external repository. Please rate, if applicable, the supporting data on the following criteria.

Is it accessible?

Yes

Is it clear?

Yes

Is it adequate?

Yes

Do you have any ethical concerns with this paper?

No

Comments to the Author

This is a well-written paper that has taken a creative approach to address the needs/assets hypothesis. This is an important concept in foraging theory but rarely addressed in marine systems. The authors set up the paper clearly in the intro and outline how they are going to address this in beaked whales. A strength of this study is how the authors first address the trade-offs between predator avoidance and foraging behavior. Next they test the hypotheses that anti-predator behavior will increase and foraging decrease when exposed to a stressor. After clearly establishing this trade off, they test the hypothesis that animals in lower body condition will forage more as predicted from the needs/assets hypotheses. The data does not support this hypothesis, but the authors provide possible explanations including a low risk life-history strategy for this species, to low variation in body condition in the 15 individuals studied, to personality impacts.

My main comment is that the paper would be improved if more information on the playbacks are included in the methods. Without carefully examining the supplemental tables it was unclear if the 3 playback animals were included in the 15. Since they are, I would make this clear in the methods and then explain how you only used the baseline data when looking at the relationship between body condition and AP/Foraging relationship (at least that is what I think you did, based on the supplemental table - if you did not do this, then you should). Overall, the methods are well-explained and appropriate. I like how the models tested are included in the supplemental material.

Minor comments:

Line 73-77: Small style recommendation - the transition between these two sentences is abrupt.

Writing would be stronger if connect ideas more clearly

Table 2: check your r value for % foraging dive vs ascent depth. The r listed is 0, yet you say this is significant.

Fig 1: One way to make it clear in the paper that some of the individuals were exposed to sonar

would be to bold those individuals in Fig 1. But if you only analyze baseline data then this is not essential (but you still need to explain this in the methods). If figure 1b some of the letters have lines through them. I am assuming this is a mistake – maybe a conversion error. If not, then you need to explain why.

Decision letter (RSPB-2021-1160.R0)

06-Jul-2021

Dear Ms Siegal:

I am writing to inform you that your manuscript RSPB-2021-1160 entitled "Beaked whales and state-dependent decision-making: how does body condition affect the starvation-predation trade-off?" has, in its current form, been rejected for publication in Proceedings B.

This action has been taken on the advice of referees, who have recommended that substantial revisions are necessary. With this in mind we would be happy to consider a resubmission, provided the comments of the referees are fully addressed. However please note that this is not a provisional acceptance.

Sincerely,
Dr Daniel Costa
mailto: proceedingsb@royalsociety.org

Associate Editor
Board Member: 1
Comments to Author:

Both myself and the reviewers thought the manuscript was well written and used a creative approach to address the needs/assets hypothesis that is rarely addressed in marine systems. Both reviewers felt the study system was well chosen and appreciated the metrics of foraging and

predator avoidance behaviors and the fairly unique ability to assess body condition in a free-ranging marine predator based on the considerable previous work done by the authors in this area. The second reviewer raised some relatively minor issues about the methods and results presentation. However, one reviewer has raised some important concerns about the results interpretation, the strength of the conclusion drawn and the influence of one outlier data point on the conclusions drawn as written. I agree with the reviewer that this issue potentially effects the suitability of the paper for Proc B and requires response or revision from the authors.

Reviewer(s)' Comments to Author:

Referee: 1

Comments to the Author(s)

I found this to be an extremely well written paper on an interesting topic. The introduction was strongly rooted in theory, and beaked whales provide an interesting case study for answering questions about state-dependence given the relative ease of calculating foraging and predator-avoidance behaviors as well as body condition. The statistical analysis was generally strong, and the authors were creative in their use of anti-predator and foraging metrics. However, I struggled with the interpretation of the main data figure (specifically, the leverage that one datapoint has on the model fit and thus interpretation and implications of the paper). I am concerned that given the cross-sectional rather than longitudinal approach, and the strong impact of a single datapoint that could easily have been affected by personality or habituation, question the strong statements underlying the pitch of the paper. As a result, I feel that the language needs to be softened throughout the paper, and that the manuscript may be better suited to a slightly lower tiered journal.

Major comments

- L26-28 suggests that skinnier animals minimized predation risk... But this is a chicken-or-egg question (what came first?). What if body condition was low because animals used more anti-predator behaviors at the expense of foraging? I don't think this necessarily suggests that the needs/assets hypothesis is not true.
- When the one animal was excluded, CLAP/CLF did not vary with body density – so statements in the abstract should be softened. Figure 1b should not have a line because it is not significant. L25-28 is not supported by the data. Without this result, the novelty of the manuscript is somewhat reduced (because, for example Figure 1A – the negative relationship between foraging and anti-predator behaviors – are an emergent property, as discussed in L310). Also, this should not be interpreted as strong negative support for the needs/assets hypothesis, as is written into several sections of the paper (e.g., L284-286).
- Much of this work, especially the discussion, is pitched as the starvation-predation trade-off. While I appreciate this effort, I think the tradeoff being measured here is foraging versus predator avoidance. The authors could add a few sentences to discuss this difference. For example, what are the body density values in this species that might lead to starvation?

Minor comments

Abstract: In abstract, response to sonar exposure is termed predator avoidance... should this be considered risk aversion? Threat avoidance? I agree it is analogous, but I do not think that behavioral changes from sonar exposure should be called predator avoidance. I appreciate the nuanced discussion section (e.g., L330-334) and would like to see more careful wording in the abstract.

L13: Start to abstract is dry

L18: grammar

L20: Where foraged, not tagged

L21: Is acoustic crypsis an anti-prey behavior as well?

L25: Simplify to lower lipid stores (remove higher body density)

L30: Abstract concluding sentence should broaden out results / discuss implications rather than justify why a certain result was not found

L65: The safety component of this should be underscored. Suggest changing to "rested more during the safety of nighttime and at deeper depths..."

L75: Suggest giving an example of anthropogenic disturbances that are concerning. Sonar?

L86: Change "this study" to "we"

L88: Suggest removing “of this marine top predator”

L90: It was unclear to me in the abstract whether sonar was experimentally added or whether it was naturally encountered. Here it is clear that experimental exposures were done – please add this to the abstract.

L124: Change grammar to “During stable five-second glide segments, measurements (acceleration...)”

L139: What is the unit for buzz rate?

L148: What constitutes “regular clicks”? Is this objective or subjective?

L149: Change to “long bottom phase durations”

L151: I found this paragraph relatively hard to follow. Perhaps a figure would help (i.e., convert the table to a figure with more helpful information).

L234: I would like to see statistical analysis in this section. Are these comparisons biologically or statistically relevant? For example, the percentage of time being non-vocal near the surface increased from 97.3 to 100%, which is a very minor change.

L213: Is there precedent for using dominant stroke frequency as an index of body size? I see in L218-219 that this is repeated, with citations, and now understand that L213 is meant to be an overview of the paragraph. I suggest removing “(an index of body size)” in L213 to resolve this confusion.

L258: I don’t believe it is necessary to specify that delta AICc is 0 here – this is clearly the best fitting model.

Fig 2: This is somewhat unclear to me – are 2013, 2015, and 2016 the three individual whales, or years, or both? Also, it would be helpful if the foraging metrics (presumably A,B,C) are somehow differentiated from the other metrics. For this figure and the others, it would be helpful to have more information about how many dives are represented.

L290: Replace “predicted” with “was related to”

L352: Replace “which annually cycle their lipid stores” with something like “which experience vast differences in lipid stores across seasons”

L377: This paragraph epitomizes my struggle with the inclusion of individual O in the regression. Because this study is cross-sectional rather than longitudinal, it seems that one whale with high (low) body condition could have simply been less of a risk taker, and has strong leverage on the linear model fit.

Referee: 2

Comments to the Author(s)

This is a well-written paper that has taken a creative approach to address the needs/assets hypothesis. This is an important concept in foraging theory but rarely addressed in marine systems. The authors set up the paper clearly in the intro and outline how they are going to address this in beaked whales. A strength of this study is how the authors first address the trade-offs between predator avoidance and foraging behavior. Next they test the hypotheses that anti-predator behavior will increase and foraging decrease when exposed to a stressor. After clearly establishing this trade off, they test the hypothesis that animals in lower body condition will forage more as predicted from the needs/assets hypotheses. The data does not support this hypothesis, but the authors provide possible explanations including a low risk life-history strategy for this species, to low variation in body condition in the 15 individuals studied, to personality impacts.

My main comment is that the paper would be improved if more information on the playbacks are included in the methods. Without carefully examining the supplemental tables it was unclear if the 3 playback animals were included in the 15. Since they are, I would make this clear in the methods and then explain how you only used the baseline data when looking at the relationship between body condition and AP/Foraging relationship (at least that is what I think you did, based on the supplemental table – if you did not do this, then you should). Overall, the methods are well-explained and appropriate. I like how the models tested are included in the supplemental material.

Minor comments:

Line 73-77: Small style recommendation – the transition between these two sentences is abrupt. Writing would be stronger if connect ideas more clearly

Table 2: check your r value for % foraging dive vs ascent depth. The r listed is 0, yet you say this is significant.

Fig 1: One way to make it clear in the paper that some of the individuals were exposed to sonar would be to bold those individuals in Fig 1. But if you only analyze baseline data then this is not essential (but you still need to explain this in the methods). If figure 1b some of the letters have lines through them. I am assuming this is a mistake – maybe a conversion error. If not, then you need to explain why.

Author's Response to Decision Letter for (RSPB-2021-1160.R0)

See Appendices A & C.

RSPB-2021-2539.R0

Review form: Reviewer 1

Recommendation

Accept as is

Scientific importance: Is the manuscript an original and important contribution to its field?

Excellent

General interest: Is the paper of sufficient general interest?

Excellent

Quality of the paper: Is the overall quality of the paper suitable?

Excellent

Is the length of the paper justified?

Yes

Should the paper be seen by a specialist statistical reviewer?

No

Do you have any concerns about statistical analyses in this paper? If so, please specify them explicitly in your report.

No

It is a condition of publication that authors make their supporting data, code and materials available - either as supplementary material or hosted in an external repository. Please rate, if applicable, the supporting data on the following criteria.

Is it accessible?

Yes

Is it clear?

Yes

Is it adequate?

Yes

Do you have any ethical concerns with this paper?

No

Comments to the Author

Thank you for your thorough responses; the manuscript has been greatly improved, and I am happy with its publication.

Decision letter (RSPB-2021-2539.R0)

10-Dec-2021

Dear Dr Siegal

I am pleased to inform you that your manuscript RSPB-2021-2539 entitled "Beaked whales and state-dependent decision-making: how does body condition affect the trade-off between foraging and predator avoidance?" has been accepted for publication in Proceedings B.

The referee(s) have recommended publication, but also suggest some minor revisions to your manuscript. Therefore, I invite you to respond to the referee(s)' comments and revise your manuscript. Because the schedule for publication is very tight, it is a condition of publication that you submit the revised version of your manuscript within 7 days. If you do not think you will be able to meet this date please let us know.

- 1) A text file of the manuscript (doc, txt, rtf or tex), including the references, tables (including captions) and figure captions. Please remove any tracked changes from the text before submission. PDF files are not an accepted format for the "Main Document".
- 2) A separate electronic file of each figure (tiff, EPS or print-quality PDF preferred). The format should be produced directly from original creation package, or original software format. PowerPoint files are not accepted.
- 3) Electronic supplementary material: this should be contained in a separate file and where possible, all ESM should be combined into a single file. All supplementary materials accompanying an accepted article will be treated as in their final form. They will be published alongside the paper on the journal website and posted on the online figshare repository. Files on

figshare will be made available approximately one week before the accompanying article so that the supplementary material can be attributed a unique DOI.

Sincerely,

Dr Daniel Costa

Reviewer(s)' Comments to Author:

Referee: 1

Comments to the Author(s).

Thank you for your thorough responses; the manuscript has been greatly improved, and I am happy with its publication.

Author's Response to Decision Letter for (RSPB-2021-2539.R0)

See Appendix D.

Decision letter (RSPB-2021-2539.R1)

20-Dec-2021

Dear Dr Siegal

I am pleased to inform you that your manuscript entitled "Beaked whales and state-dependent decision-making: how does body condition affect the trade-off between foraging and predator avoidance?" has been accepted for publication in Proceedings B.

Data Accessibility section

Open Access

Paper charges

You are allowed to post any version of your manuscript on a personal website, repository or preprint server. However, the work remains under media embargo and you should not discuss it

with the press until the date of publication. Please visit <https://royalsociety.org/journals/ethics-policies/media-embargo> for more information.

Sincerely,
Editor, Proceedings B
mailto: proceedingsb@royalsociety.org

Appendix A

Re: Resubmission of RSPB-2021-1160

Dear Professor Costa,

Thank you very much for inviting us to resubmit our manuscript (RSPB-2021-1160) to the Proceedings of the Royal Society B. We would also like to thank both anonymous reviewers for their highly constructive feedback. Please see the corresponding response documents for how we have addressed their comments, which have considerably improved the manuscript.

Reviewer 1's primary concern was regarding the interpretation of the main data figure, in particular the strong impact of one datapoint and the cross-sectional nature of our study. We agree that the cross-sectional approach means that the statistically significant positive result could be due to the particularities of the individual whales sampled (via tagging), and that the single influential datapoint may have been affected by other factors (e.g., personality or immediate factors such as sensitisation to prior sonar exposure), so it is fair to question the generalisability of that result. We have therefore reduced emphasis on the positive correlation found in Figure 1b.

However, while we agree that it is not justified to argue that the unexpected positive relationship between body density and CI_{AP}/CI_F can be generalized beyond our study, it is clear that our study did not support the expected negative relationship between body density and CI_{AP}/CI_F as *a priori* predicted from the needs/assets hypothesis. This finding remains an important and key result, indicating that individuals in worse condition might not increase foraging and individuals in better condition might not increase avoidance behaviour as expected. This finding, in addition to our study establishing effective foraging and anti-predator indicators in a beaked whale, which were modified as expected by experimental exposure to sonar, will be of broad general interest to animal biologists. We therefore hope you will find the revised manuscript acceptable for Proc Roy Soc B.

We have provided a 'clean' word document and a word document with changes tracked from the previously submitted version.

Sincerely,

Eilidh Siegal, on behalf of all co-authors

Appendix B

Response to Reviewer 1 (resubmission of RSPB-2021-1160)

Please note line numbers refer to the track changes version of the manuscript

Comment number	Reviewer 1 – major comments	Response
1	I found this to be an extremely well written paper on an interesting topic. The introduction was strongly rooted in theory, and beaked whales provide an interesting case study for answering questions about state-dependence given the relative ease of calculating foraging and predator-avoidance behaviors as well as body condition. The statistical analysis was generally strong, and the authors were creative in their use of anti-predator and foraging metrics.	Thank you very much for your review of our manuscript and for providing positive, constructive and useful feedback. We really appreciate your thorough reading and detailed comments, which have helped improve the manuscript and are addressed in the table below.
2	However, I struggled with the interpretation of the main data figure (specifically, the leverage that one datapoint has on the model fit and thus interpretation and implications of the paper). I am concerned that given the cross-sectional rather than longitudinal approach, and the strong impact of a single datapoint that could easily have been affected by personality or habituation, question the strong statements underlying the pitch of the paper. As a result, I feel that the language needs to be softened throughout the paper, and that the manuscript may be better suited to a slightly lower tiered journal.	We agree that the one datapoint had a good deal of leverage on the results. We have made this clear in the text and figure, and have modified the manuscript to not base our conclusions on this. In fact, the empirical lack of the predicted relationship between body density and CI_{AP}/CI_F remains a valuable result. Few studies have assessed foraging-predator avoidance trade-offs in marine top predators, and even fewer studies have looked at how these trade-offs could relate to body condition. As such, the empirical evidence in our study that a pattern expected by the needs/assets hypothesis did not occur provides novel insights into behavioural trade-offs and how these trade-offs might relate to body condition in a difficult-to-study marine animal, and thus this study (which uses non-invasive widely-applicable methods) still has general interest to animal biologists. We have modified the language throughout the manuscript as suggested. For example, instead of focussing on the positive line in Figure 1b, the text now focuses on the relationship between body condition and foraging-predator avoidance trade-offs not being that as predicted by the needs/assets hypothesis (i.e., the results clearly don't indicate a negative relationship) and the potential reasons for why this might be (e.g., body condition values were not extreme enough to drive increased risk-taking etc.). The following are examples of where the language has been changed and softened:

		 • Results, section c [L297 - 313]. • Discussion first paragraph [L327 – 330] • Discussion section c (e.g. L387 – 405; 426 - 428; 434 - 443; 452 - 459] • Conclusions [L501 -518]
3	L26-28 suggests that skinnier animals minimized predation risk... But this is a chicken-or-egg question (what came first?). What if body condition was low because animals used more anti-predator behaviors at the expense of foraging? I don't think this necessarily suggests that the needs/assets hypothesis is not true.	We agree that this could be a chicken-or-egg question, which we had stressed in our discussion on how factors such as personality could lead to the observed data. The wording in the abstract has been amended accordingly [L30 – 34], so as to not state that animals in worse condition minimised predation risk (or used more anti-predator behaviours at the expense of foraging). The wording instead now focusses on reflecting the result and consistency with comment #4 below. We also agree that the relationship between body density and Cl_{AP}/Cl_F does not necessarily suggest the needs/assets hypothesis is not true and have addressed this (please see response to comment #4 for more detail).
4	When the one animal was excluded, Cl_{AP}/Cl_F did not vary with body density – so statements in the abstract should be softened. Figure 1b should not have a line because it is not significant. L25-28 is not supported by the data. Without this result, the novelty of the manuscript is somewhat reduced (because, for example Figure 1A – the negative relationship between foraging and anti-predator behaviors – are an emergent property, as discussed in L310). Also, this should not be interpreted as strong negative support for the needs/assets hypothesis, as is written into several sections of the paper (e.g., L284-286).	Thank you very much for this constructive feedback. We have modified the language throughout the manuscript (please see response to comment #2). We have not removed the model line from Figure 1b [L728] as the result is supported by the empirical data and there is no reason to doubt the validity of that datapoint. However, we have softened our interpretation of the generalizability of the surprising positive relationship as we agree that the cross-sectional nature of the study means that the statistically significant positive result could have been due to the particularities of which whales were sampled (via tagging) in our study. We have edited L25 – 28 [now L30 – 34] accordingly. We agree that the lack of relationship between body density and Cl_{AP}/Cl_F does not equate to there being strong negative support for the needs/assets hypothesis and appreciate that our wording on this was too strong. Figure 1b convincingly demonstrates a lack of the expected negative relationship between body density and Cl_{AP}/Cl_F as predicted from

		the needs/assets hypothesis (i.e., that animals in worse condition have higher levels of foraging). We have modified the wording throughout (including L284-286, now L327-330), to state that the results are “unexpected” (rather than “opposite” or “contrary”) from the predictions of the needs/assets hypothesis (but we note the lack of proof that the hypothesis is not true).
5	Much of this work, especially the discussion, is pitched as the starvation-predation trade-off. While I appreciate this effort, I think the tradeoff being measured here is foraging versus predator avoidance. The authors could add a few sentences to discuss this difference. For example, what are the body density values in this species that might lead to starvation?	Thank you for raising this point. We do not have a way to estimate what body density might relate to starvation in this species. However, we do agree with the reviewer, as indicated in the discussion that the predicted trade-off may operate at the extreme levels of body condition that reflect risk of starvation. We therefore accept the point that “foraging versus predator avoidance” more accurately describes the trade-offs being measured, and have changed the wording throughout the manuscript to “foraging versus predator avoidance trade-off” (rather than “starvation-predation trade-off”). We have included text describing the difference between the two (L47 - 57).
Comment number	Reviewer 1 – minor comments	Response
6	Abstract: In abstract, response to sonar exposure is termed predator avoidance... should this be considered risk aversion? Threat avoidance? I agree it is analogous, but I do not think that behavioral changes from sonar exposure should be called predator avoidance. I appreciate the nuanced discussion section (e.g., L330-334) and would like to see more careful wording in the abstract.	Thank you for highlighting this. We agree that behavioural changes to sonar shouldn't be termed “predator avoidance”. The text has therefore been updated to “increased risk aversion” [L29].
7	L13: Start to abstract is dry	The start of the abstract has been modified to draw the reader in [L13-15].
8	L18: grammar	Sentenced amended to improve grammar [L19-22]

9	L20: Where foraged, not tagged	We reworded this to add that the animals were foraging in the study area [L23]
10	L21: Is acoustic crypsis an anti-prey behavior as well?	Yes, acoustic crypsis (silence during ascents from foraging dives and whilst at the surface) are likely anti-predator behaviours of beaked whales (Soto et al., 2020). We have added a definition to the abstract to clarify this [L26].
11	L25: Simplify to lower lipid stores (remove higher body density)	We have deleted “higher body density” [L30].
12	L30: Abstract concluding sentence should broaden out results / discuss implications rather than justify why a certain result was not found	As suggested, we have altered the last sentence to broaden out the results and implications [L37 - 40].
13	L65: The safety component of this should be underscored. Suggest changing to “rested more during the safety of nighttime and at deeper depths...”	We have updated this sentence in line with the suggestion (changed from “rested more during night-time and at deeper depths” to “rested more during the safety of night-time and at deeper depths”) [L83]
14	L75: Suggest giving an example of anthropogenic disturbances that are concerning. Sonar?	This sentence has been updated to include the example of naval sonar as a concerning anthropogenic disturbance [L97]
15	L86: Change “this study” to “we”	Changed as suggested [L108]
16	L88: Suggest removing “of this marine top predator”	Changed as suggested [L110]
17	L90: It was unclear to me in the abstract whether sonar was experimentally added or whether it was naturally encountered. Here it is clear that experimental exposures were done – please add this to the abstract.	The abstract has been updated to clarify that that sonar exposures were experimental [L28]
18	L124: Change grammar to “During stable five-second glide segments, measurements (acceleration...)”	Changed as suggested [L157]
19	L139: What is the unit for buzz rate?	The unit for buzz rate (buzz h ⁻¹) have been added [L173]
20	L148: What constitutes “regular clicks”? Is this objective or subjective?	A definition of regular clicks has now been provided [L135 – 138], and a sentence [L141 - 143] added to address the issue of potential subjectivity in the identification of regular clicks (and buzzes).
21	L149: Change to “long bottom phase durations”	Changed as suggested [L183]

22	L151: I found this paragraph relatively hard to follow. Perhaps a figure would help (i.e., convert the table to a figure with more helpful information).	Yes – this paragraph was rather difficult to follow (not helped by our omission of numbering). We have improved and corrected this – and hope that this is now sufficiently clear. We have added numbering for all indicators [L187 – 200] and have reorganised Table 1 so as to follow this number order. Other reviewers commented on the utility of Table 1 and so we prefer to retain this if possible. Though we tried, we could not develop a figure to effectively convey this complex set of behavioural indicators.
23	L234: I would like to see statistical analysis in this section. Are these comparisons biologically or statistically relevant? For example, the percentage of time being non-vocal near the surface increased from 97.3 to 100%, which is a very minor change.	This section [L272] provides support for our foraging and anti-predation indicators, in terms of expected changes with an increase in perceived risk. With only three tag deployments, we are not keen to attempt statistical analyses, but consider this section nevertheless valuable in showing consistency of response between the three individuals.
24	L213: Is there precedent for using dominant stroke frequency as an index of body size? I see in L218-219 that this is repeated, with citations, and now understand that L213 is meant to be an overview of the paragraph. I suggest removing “(an index of body size”) in L213 to resolve this confusion.	Yes, fundamental stroke frequency was originally related to body size across species [44] but has also been applied to proxy body size within a species (e.g., in pilot whales, Isojunno et al. 2018). Changed as suggested [L249].
25	L258: I don’t believe it is necessary to specify that delta AICc is 0 here – this is clearly the best fitting model.	We have deleted “($\Delta AICc = 0$, Akaike weight = 0.48)” accordingly in L296 - 297. These values can also be found in Supplement 4, which is referenced in this sentence.
26	Fig 2: This is somewhat unclear to me – are 2013, 2015, and 2016 the three individual whales, or years, or both? Also, it would be helpful if the foraging metrics (presumably A,B,C) are somehow differentiated from the other metrics. For this figure and the others, it would be helpful to have more information about how many dives are represented.	We have edited Figure 2:  • The previous labels of “2013”, “2015” and “2016” represented three individual whales tagged in those years. These labels have now been changed to “a”, “l” and “n” (the individual whale codes used in Figure 1 and Table S1). • The plot layout has been changed (previously arranged as a 3 X 3 matrix of subplots), so that the foraging indicators are on the top row, and anti-predator indicators are on the bottom row (and labelled accordingly on the outer Y axis). • For the three indicators (ascent-pitch-shalowness, ascent depth, and ascent-straightness index) that are calculated as mean values across mid-depth and long-deep dives, the number of dives is now represented as a label above the bar.

		These changes have been explained in the Figure description [L743 - 747].
27	L290: Replace “predicted” with “was related to”	Changed as suggested [L335]
28	L352: Replace “which annually cycle their lipid stores” with something like “which experience vast differences in lipid stores across seasons”	Changed as suggested, replacing “which annually cycle their lipid stores” with “which, with seasonal migrations, experience vast differences in their lipid stores across seasons” [L410 - 411]
29	L377: This paragraph epitomizes my struggle with the inclusion of individual O in the regression. Because this study is cross-sectional rather than longitudinal, it seems that one whale with high (low) body condition could have simply been less of a risk taker, and has strong leverage on the linear model fit.	The wording of this paragraph and the following paragraph [L444 - 476] has been amended in-line with comments #2 and #4 above (wording softened, and discussion of the nature of this study being cross-sectional rather than longitudinal). We also consider that individual O could have simply been more of a risk taker.

References

Aguilar de Soto N., Visser F., Tyack P. L., Alcazar J., Ruxton G., Arranz P., Madsen P. T. and Johnson M. (2020). Fear of killer whales drives extreme synchrony in deep diving beaked whales. *Scientific Reports*, 10, 1-9.

Isojunno, S., Aoki, K., Curé, C., Kvadsheim, P. H. and Miller, P. J. O. M. (2018). Breathing patterns indicate cost of exercise during diving and response to experimental sound exposures in long-finned pilot whales. *Frontiers in Physiology*, 9, 1462.

Appendix C

Response to Reviewer 2 (resubmission of RSPB-2021-1160)

Please note line numbers refer to the track changes version of the manuscript

Comment number	Reviewer 1 – major comments	Response
1	This is a well-written paper that has taken a creative approach to address the needs/assets hypothesis. This is an important concept in foraging theory but rarely addressed in marine systems. The authors set up the paper clearly in the intro and outline how they are going to address this in beaked whales. A strength of this study is how the authors first address the trade-offs between predator avoidance and foraging behavior. Next they test the hypotheses that anti-predator behavior will increase and foraging decrease when exposed to a stressor. After clearly establishing this trade off, they test the hypothesis that animals in lower body condition will forage more as predicted from the needs/assets hypotheses. The data does not support this hypothesis, but the authors provide possible explanations including a low risk life-history strategy for this species, to low variation in body condition in the 15 individuals studied, to personality impacts.	Thank you very much for your positive feedback and review. We really appreciate your time, careful reading of the manuscript, and constructive feedback.
2	My main comment is that the paper would be improved if more information on the playbacks are included in the methods. Without carefully examining the supplemental tables it was unclear if the 3 playback animals were included in the 15. Since they are, I would make this clear in the methods and then explain how you only used the baseline data when looking at the relationship between body condition and AP/Foraging relationship (at least that is what I think you did, based on the supplemental table – if you did not do this, then you should). Overall, the methods are well-explained and appropriate. I like how the models tested are included in the supplemental material.	Text has been added to the method (L144 – 149) to clarify that: • The 3 playback animals were included in the 15;• Only baseline data were used when assessing the foraging-predator avoidance trade-off and the relationship of this trade-off with body condition; and• Post-exposure data were only used to test whether the behavioural indicators were involved in trade-offs based on an increase in perceived risk (i.e., sonar exposure).

Comment number	Reviewer 2 – minor comments	Response
3	Line 73-77: Small style recommendation – the transition between these two sentences is abrupt. Writing would be stronger if connect ideas more clearly	We have added a sentence [L92 – 96] in between these two sentences to connect the ideas, so the transition is less abrupt.
4	Table 2: check your r value for % foraging dive vs ascent depth. The r listed is 0, yet you say this is significant.	Our apologies - the Spearman's rank correlation value in Table 2 (for % foraging dive vs ascent depth) was correct (= 0.00); however, this was incorrectly identified as significant (labelled '*'). This has been corrected in Table 2 (i.e., this relationship is no longer identified as significant). This relationship was not mentioned elsewhere in the manuscript.
5	Fig 1: One way to make it clear in the paper that some of the individuals were exposed to sonar would be to bold those individuals in Fig 1. But if you only analyze baseline data then this is not essential (but you still need to explain this in the methods).	Only baseline data were analysed for the three individuals exposed to sonar (except when specifically addressing the effects of sonar exposure on the composite indices). Text has been added to the Figure 1 caption to clarify this [L733 – 734]
6	If figure 1b some of the letters have lines through them. I am assuming this is a mistake – maybe a conversion error. If not, then you need to explain why.	Thank you for pointing this out. The horizontal line through the letters in Figure 1b represents the 95% posterior credible interval around the body density estimates. Text has been added to the Figure 1 caption to clarify this [L731]

Appendix D

Re. Acceptance of RSPB-2021-2539

Dear Professor Costa and Referees,

Thank you very much for accepting our manuscript (RSPB-2021-2539) for publication in Proceedings of the Royal Society B.

We are absolutely delighted, and would like to sincerely thank you for your time and for the reviewers' comments and feedback, which greatly improved the manuscript.

We have provided a 'clean' word document and a word document with changes tracked from the previously submitted version.

The changes to the manuscript made since the previous resubmission are:

- Updated a contact email address
- Added a DOI link to the supporting dataset on Dryad under the 'Data accessibility' statement. The dataset is currently unpublished but can be accessed through a temporary link: <https://datadryad.org/stash/share/YgeJyYBldj8AND-kBGeS3SZUb4pvTqPWBVHxtZgYziU>
- Small grammatical (typo) corrections
- L249-250, L265, L277, L710 – presented results to 1 decimal place instead of 2 decimal places for consistency within the manuscript and supplementary materials (line numbers are for the track changes version of the manuscript).
- Table 1 – Added a further reference for evidence of bounce dives in *Mesoplodon densirostris*
- Table 2 – Due to a clerical error, the maximum value for silent dives was mistakenly given as 87.3% in Table 2. This does not impact any other results and the correct value (86.3%) was quoted elsewhere in the manuscript (L285 only).
- Table 2 - Due to a clerical error, the correlation value between buzz rate and ascent-pitch-shalowness was mistakenly given as -0.53 (instead of -0.54) and an asterisk representing a significant relationship ($p < 0.05$) between these two indicators was mistakenly left out of Table 2. The value of the correlation between buzz rate and ascent-pitch-shalowness (now noted as -0.54 instead of -0.53) remains little changed, and the relationship is now noted as significant at the $p < 0.05$ level. The statistical significance of the relationship does not affect any conclusions as the negative correlation itself was already incorporated within the results section (L243 – 244: "Foraging indicators were generally positively correlated with each other, and negatively correlated with anti-predator behaviour"). All results have been thoroughly re-checked with no other changes.
- Reduced the total number of references (from 64 to 58) whilst keeping references appropriate
- Updated reference list where the full author list was not previously given

Please don't hesitate to get in touch if any further information is required. Many thanks again for your time.

Sincerely,

Eilidh Siegal, on behalf of all co-authors